# Willingness of the New Generation of Farmers to Participate in Rural Tourism: The Role of Perceived Impacts and Sense of Place

**Binbin Li, Zengyu Mi * and Zhenghe Zhang**

College of Economics and Management, China Agricultural University (CAU), Beijing 100083, China; libinbin01234@163.com (B.L.); zhangzhenghe@cau.edu.cn (Z.Z.)

* Correspondence: zymi@cau.edu.cn; Tel.: +86-1881-302-6998

**Abstract:** This study aims to assess the willingness of the new generation of farmers in China to return to their hometowns for the purpose of contributing to rural tourism. Based on the hierarchy of effects models (HOE) and the social representation theory (SRT), this study utilized an analysis model of perceived rural tourism impacts (P), sense of place (SoP), and willingness to participate (WTP) by the new generation of farmers, where sense of place was deconstructed into the two independent dimensions of place identification (PI) and place attachment (PA). A total of 263 valid questionnaire surveys were collected in Yanling County, a demonstration county for leisure agriculture and rural tourism in China. The structural equation modeling (SEM) technique was used to analyze the data. The results show that the perceived impacts of rural tourism on the new generation of farmers had a significant positive impact on their willingness to participate in the development of rural tourism. It was also found that sense of place played a mediating role in the perceived impacts of rural tourism and willingness to participate. Additionally, perceived impacts of rural tourism had a significant positive effect on place identity, where place identity played a mediating role between perceived impacts of rural tourism and place attachment. Place identity had a significant positive effect on place attachment, and place attachment played a mediating role between place identity and willingness to participate. The practical implications of these findings for future research and rural tourism development are also discussed.

**Keywords:** rural tourism; sense of place; willingness to participate; new generation of farmers; China

## 1. Introduction

As rural areas have gradually become marginalized during the process of urbanization, rural decline has become a major challenge for developing countries in the 21st century. As an important driving force for rural development transformation, rural tourism has been recognized as a feasible approach to rural revitalization in China [1], this being especially the case for remote rural areas that lack resources and land (for example, rural areas in Guangxi contain mostly mountains and rivers, lacking arable land; rural desertification in Gansu is serious, due to lack of water resources). After nearly three decades of development, China's rural tourism initiative has achieved some success. However, with large numbers of second-generation groups entering big cities in China and rural aging becoming a serious issue, rural tourism faces the problem of who will take over. It seems necessary to understand the willingness of the new generation of farmers to participate in the development of rural tourism, which determines the intergenerational replacement and sustainability of the development of rural tourism in China.

Rural tourism has changed residents' perception of the countryside and themselves, generating great interest among the second generation of farmers. On the one hand, rural tourism transforms

farmers' living space into consumption space for urban residents, while the productive space is also transformed into consumption space [2]. With the development of rural tourism and its success in some areas, the residents' perceptions of rural areas have begun to be differentiated between those who describe rural areas as having been abandoned and disadvantaged, and those who present rural areas as spaces for economic activity, mainly related to tourism activities [3]. On the other hand, the development of rural tourism has changed the identity of people in rural China and diversified management has made farmers perceive themselves as both entrepreneurs and producers [4]. Moreover, the development of rural tourism has also changed the perception of rural residents, clearly increasing their available opportunities and contributing significantly to making rural areas more attractive to young adults [5].

Over the past 30 years, extensive research has been conducted concerning residents' willingness to participate in the development of rural tourism (e.g., Almeida-García et al. [6], Robin and Gursoy [7], Garcia et al. [8], Rivera et al. [9], Zhang and Zhao [10], Stylidis et al. [11], Stylidis [12]). However, these studies mainly focused on the analysis of the participation of local residents, and the research subject has not yet been extended to the second generation of farmers. In the context of an aging workforce, this does not seem to be an effective solution to intergenerational replacement and the sustainability of rural tourism. In addition, the existing research studies mainly use the analytical framework of social exchange theory, which is not applicable to rural second-generation groups that are not directly involved in rural tourism. This is because they are not directly involved in the social exchange system of rural tourism. Therefore, it is necessary to adopt social representation theory. As direct participants in rural tourism, second-generation farmers will generate a series of cognitive, emotional, and other psychological responses under the condition of successful rural tourism. The hierarchy of effects model provides an ideal perspective for us to study the willingness of the new generation of farmers to participate in rural tourism development through the segmentation of individual psychological reactions. Based on social representation theory, the hierarchy of effects model, and sense of place theory, this paper intends to construct an analytical framework for the willingness of rural second-generation groups to participate in rural tourism development, including the perceived influence of rural tourism, sense of place, and willingness to participate; and analyze the influence mechanism of sense of place in the willingness to participate. Through this study, individuals will gain a new understanding of the intergenerational change of rural tourism.

Hence, the main purpose of this study was to analyze the influence of perceived rural tourism impacts on the willingness towards rural tourism from the perspective of sense of place. Specifically, it examines the direct and indirect effects of perceived rural tourism impacts and willingness to participate, and explores the mediation effect of sense of place. The findings of this study will help to improve the quality and efficiency of rural tourism, and provide a new perspective for Chinese government to develop rural tourism. This paper is composed of six sections and organized as follows. Following this introductory section, Section 2 reviews the theoretical background and literature. The research hypotheses are proposed on the basis of theoretical analysis and literature review. Section 3 describes the study methodology. Section 4 analyzes the data obtained from the investigation and presents the results. Section 5 discusses the results and their implications for future research and practice. Section 6 provides the conclusions to the study.

## 2. Theoretical Background and Study Hypotheses

In this section, the background for the topic is provided. Then, the relationship between the perceived impacts of rural tourism, sense of place, and willingness to participate are reviewed through literature. Finally, we develop our conceptual model and propose the research hypotheses.

*2.1. Theoretical Background*

2.1.1. Hierarchy of Effects Model

The hierarchy of effects (HOE) model reveals the process by which individuals form attitudes and aspirations towards a product or thing. The model shows that under external stimuli, individuals have three stages of consecutive psychological reaction: cognition, emotion, and willingness [10]. Smith et al. [13] further refined the HOE model as having five steps across three phases: the cognitive phase, emotional phase, and willingness phase. The cognitive stage includes three steps of awareness, learning, and acceptance/rejection; the emotional stage includes the product preference step; and the willingness stage includes the behavioral willingness step. The HOE model is widely used to explain the areas of participation, willingness, psychology, team loyalty, and communication [10,14–18].

The previous application of HOE models to rural tourism have mainly focused on the analysis of the willingness of tourists [19–21], while in recent years, these models have begun to be used to analyze the willingness of community residents to participate [10]. Stimulated by the success of rural tourism, the second generation of farmers, as indirect participants in rural tourism, produces a series of psychological reactions, such as cognition, emotion, and willingness. Based on the analysis above, this study introduces HOE models into framework analysis, which will help to clarify the perception mechanism of the new generation of farmers' towards their participation in the development of rural tourism. On this basis, the main research framework of this paper is constructed, involving perceived impacts of rural tourism, sense of place, and willingness to participate.

2.1.2. Social Representation Theory

Social representation theory (SRT) provides a strong theoretical framework for promoting community participation in rural tourism development and in achieving the sustainable development of rural tourism. Moscovici defines "social representation" as "a system of knowledge of various expectations, images, and values, which have their own cultural meaning and are independent of individual experience" [22]. The formation of social representation mainly comes from publications and electronic media, social interactions, and direct experience [23], and the resulting social consensus will directly affect people's attitudes towards a certain thing. For the new generation of Chinese farmers, their working areas are separated from their hometowns, so they are not directly affected by rural tourism. Their perceptions and attitudes towards rural areas and rural tourism are mainly influenced by group imagery. They live in an information-based society and communicate more closely and rapidly with their peers through the Internet. Therefore, it is necessary to adopt the SRT to incorporate group imagery into the cognitive analysis of rural tourism. Through analysis using SRT, we incorporate social consciousness into the perception of the impact of rural tourism by the second generation of agricultural workers. On the basis of the work by Pearce [23] and SRT, combined with the development of rural tourism itself, the formation mechanism of rural tourism social representation is shown in Figure 1. The success of rural tourism affects the new generation of farmers through direct experiences, social interactions, and media propaganda, thus forming an individual perception of rural tourism—so-called social representation. This social representation will guide and control an individual's behavioral response to the impact of tourism, and the consequences of its actions, in turn, will modify the individual's original social representation of tourism.

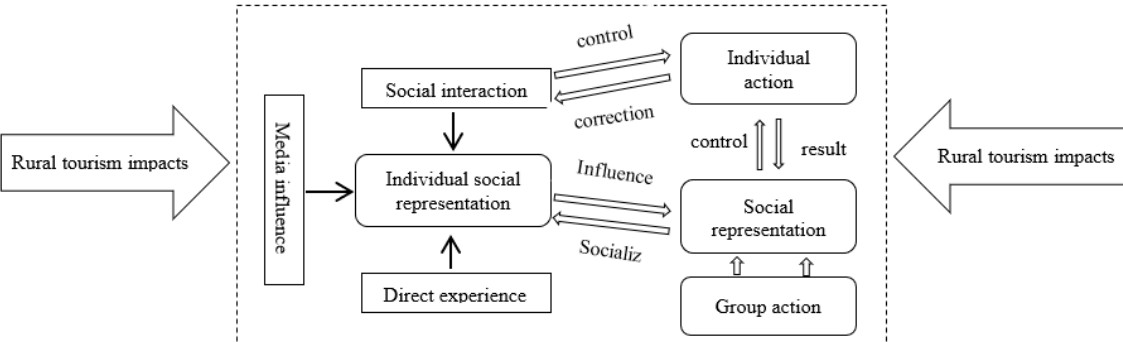

**Figure 1.** Social representation mechanism of rural tourism (modified based on Ying [24]).

### 2.1.3. Sense of Place

Sense of place (SoP) reflects the connection between people's emotions and places, and is a special relationship between people and the land transformed through cultural and social characteristics [25]. Since the second generation of farmers grew up in the countryside in the early years, their perceptions of SoP, daily experience, and social relations will affect their SoP. The rural landscape and lifestyle become their sustenance to the rural emotional attachment. The new generation of farmers were born and lived in rural areas in their early years. They gained psychological dependence and identity from their hometown, even though they worked and lived in the city.

Studies have shown that SoP is mainly composed of two dimensions: "place identity" and "place attachment" [26]. Place identity reflects an individual's emotional attachment to a place [27]. Place attachment is the intensity of an individual's perceptions of the connection between himself or herself and the place [28]. This reflects the degree of attachment of the individual to the local function when the individual is active; that is, the irreplaceability of the place in completing the subject activity [29]. Place attachment and place identity are two independent concepts. Place identity expresses the individual's emotional attachment to the place, while place attachment expresses the individual's functional identity related to the place. Dividing the SoP into place attachment and place identity is conducive to clarifying the logical relationship between the two concepts.

### 2.2. Literature Review and Study Hypothesis

### 2.2.1. Perceived Impacts of Rural Tourism and Residents' Willingness to Participate

The impacts of tourism on the economy, society, culture, and the environment of destinations has been widely recognized [6,30–32]. As community residents are recognized as an essential ingredient of the "hospitality atmosphere" at a destination, it is necessary for residents to participate in tourism planning [33]. The perceived impacts of tourism determine residents' attitudes towards tourism [34,35]. The perceptions and attitudes of community residents towards the development of rural tourism is an important factor in the successful development, marketing, and operations of existing and future tourism projects [36]. Tourism development can lead to changes in the livelihoods of the residents, as well as their lifestyles, social networks, and living environments, which may all have an impact on their perceptions of being a rural resident and living a rural life [37–39].

Extensive research concerning residents' perceived impacts of tourism and residents' willingness to participate in tourism already exists [35,40,41]. From the perspective of tourism impact perception, residents' perception of tourism can be divided into economic, social, cultural, and environmental aspects [11]. Chen's [42] analysis of Taiwanese heritage tourism revealed that the economic impact perceived by residents can significantly affect the support for tourism development compared with other impacts. Sinclair-Maragh et al. [43] divided residents into different groups according to their attention to tourism development related to economy, society, culture, the environment, and public services. The study found that people with different levels of attention had different tourism perceptions, and the

focus of attention affected their willingness to participate in rural tourism. From the perspective of the actual effect, the perception of tourism impact can be divided into positive and negative effects. Positive effects include promotion of employment, increase of income, and improvement of environment, while negative effects include congestion brought by tourism development and destruction to quiet rural environments [6]. Segota et al.'s [44] study of the Slovenian lake and mountain destination of Bled found that residents with a high degree of knowledge had a higher degree of participation and a more positive perception of tourism than all the other groups, and those with a low degree of knowledge had a low degree of participation and a negative perception of tourism. Essentially, cognitive differences, in turn, affect participation in tourism.

For the new generation of farmers, as they do not work in the countryside, they are not directly involved in the development of rural tourism. However, they have obtained an understanding of the status quo of rural tourism development from relatives and friends or other public media, and this will affect their willingness to participate in rural tourism. Especially for the regions in China where rural tourism development is more successful, and with the continuing development of tourism in these rural areas, the positive impact of rural tourism on migrant workers will gradually increase. All of this will have a positive impact on these individuals returning (through business or investment) to these areas. Previous studies focused on the new generation of migrant workers show that when urban–rural economic differences gradually narrow, they are more likely to return to their hometowns [45,46].

Based on the theoretical and empirical analysis above, hypotheses was proposed as follows:

**Hypothesis 1 (H1).** *There is a direct positive relationship between the perceived impacts of rural tourism and residents' willingness to participate in tourism activities.*

### 2.2.2. Perceived Impacts of Rural Tourism and Sense of Place

The impacts of tourism perceived by the residents will directly affect the image of the tourism destination, and this in turn will affect the SoP of the residents. The tourism destination image refers to the perception and emotional evaluation of tourists based on the various attributes of tourism destinations. Prayag and Ryan [47] have shown through research on Mauritius that the image of a tourism destination plays an important role in cultivating the attachment of tourist destinations. The image of a tourism destination is an important source of tourism attachment. Therefore, we believe that the perceived impacts of tourism play an important role in either cultivating and strengthening or weakening the SoP amongst tourism residents.

Residents influence environmental attachment and environmental behavior by perceiving the special meaning or value of tourism to the living environment. Therefore, the perceived impacts of tourism can be used as a pre-variable to describe its influence on the place attachment and place identity of the residents in a tourist destination. If the residents perceive the value or meaning of tourism development as being positive for the environment in which they live, they will have a positive emotional connection with the place [48]. That is, residents that perceive the impact of tourism as being positive experience a positive impact on their place attachment and place identity, and vice versa.

The study by Du and Su [49] of rural tourism in Anji, Zhejiang, found that perceived impacts of tourism by residents had a direct impact on their sense of community. Abdollahzadeh and Sharifzadeh's [50] research reached similar conclusions, stating that the positive influence of tourism perception will contribute to the ascension of place attachment. Long term residents and those born locally are more sensitive to the impacts of tourism and attachment to the community when tourism develops [8].

Some literature studies included sense of place as a variable of external impact in the analytical framework of the model [51,52]. Li's [53] research on Sanqing mountain in Jiangxi province further suggested that perceived impacts of rural tourism had a feedback effect on sense of place. For the new generation of farmers, the sense of place—as the inherent connection between the new generation

of farmers and the place—is a dynamic variable. In the analysis framework of the hierarchy of effects (HOE) model, cognition is the preorder variable of emotion, and the influence of rural tourism cannot directly affect the sense of place, but needs to affect the influence of rural tourism through perception [10].

Based on the above analysis, we propose the following hypothesis.

**Hypothesis 2 (H2).** *There is a direct positive relationship between the perceived impacts of rural tourism and SoP.*

### 2.2.3. Sense of Place and Residents' Willingness to Participate

Studies have shown that place attachment and place identity are closely related to residents' social development expectations, satisfaction, enthusiasm about community development, and support for the development of public facilities [54–56]. From the perspective of the attachment between people and places, place identity is the emotional attachment between people and place, and place attachment is the functional attachment between people and place. Attachment to place is economically, socially, and culturally oriented, and is pervasive in everyday experiences of a place and in memories of place being recalled [57].

SoP and community participation are key factors affecting the sustainable development of rural tourism. The relationship between sense of place and residents' willingness to participate has been tested in many places. Gursoy and Rutherford [58] built a model of travel support with multiple elements, using the analysis of the two-stage structural equation model. It can be concluded that in addition to perceived benefits and other factors, place identity will affect supporting behavior. Nicholas et al.'s [59] study on Pitons Management Area (PMA), a world heritage site, had the same view as Nicholas et al. A study of the residents of the seven valley wetlands in southwestern Taiwan found that the increase of community attachment had a direct impact on the residents' participation in the development of rural tourism [60]. In a separate study on small tourism business owners in South Australia, it was found that the place attachment of tourism business owners had a significantly positive effect on their support for the community, which in turn has a significantly positive effect on enterprise performance [61].

SoP represents a positive connection or bond between a person and rural place, thus individuals in a community connect personal glory with local glory. Ngo and Brklacich's [62] research on southern Ontario, Canada, found that in the context of counter-urbanization, new farmers would spontaneously extract and construct a sense of place from the natural and social landscape of urban and rural environments, and the constructed sense of place would help them establish a longer relationship with the countryside. In the process of anti-urbanization, the second generation of farmers consciously constructed their own sense of place from the memory of the city and the countryside, with which they established a connection. This connection connected the local honor with their own honor, and the enhancement of the sense of place contribute to their participation in rural activities.

Based on the above analysis, we propose the following hypothesis.

**Hypothesis 3 (H3).** *There is a direct positive relationship between SoP and residents' willingness to participate in rural tourism.*

### 2.2.4. Mediation Effect of Sense of Place on the Perceived Impacts of Residents' Willingness to Participate

There are many variables that play a mediating role between perceived impacts of rural tourism and willingness to participate, such as the degree of residents' participation and their attitudes [10,41,43], but there are few studies that have included SoP as a mediator. The mediator variable is considered to be caused by a free variable and affects the change in the dependent variable [63]. The perceived impacts of rural tourism have a direct impact on the SoP, and the SoP has a direct impact on residents'

willingness to participate, thus developing a path of influence with SoP as the mediator: perceived impacts of rural tourism contribute to sense of place, which contributes to willingness to participate. As a kind of external environment perception variable, perceived impacts of rural tourism directly affect the positive and negative impacts of SoP. This kind of SoP will be enhanced or weakened, and this will directly affect the new generation of farmers by encouraging them to return to their hometowns for business or investment.

Place attachment reflects functional attachment and refers to the ability of a place to meet the physical and psychological needs of the individual. Place attachment refers to the connection between a person and the environment to which they are exposed [57]. Place attachment is a form of connection between a person and an environmental setting [57]. Place identity focuses on the symbolic meaning of a place and the human being, and is the belief, emotional belonging, and values based on the understanding of the relationship between the human and the earth. It is the personal identity formed by the interaction between the human and the physical environment. Therefore, the formation of place identity usually takes a long time. Moore and Graefe [64] showed that place dependence positively affects place identity, and repeated visits to tourist attractions due to tourists' attachment to that destination may generate tourism identity. Tang et al. [65] and Shamsuddin [66] provided empirical evidence indicating that residents' attachment to local areas directly affects the identity of the residents. As previously described, identity was affected by attachment. Attachment, therefore, is a process of forming individual and group identities [67] in such a way that it appears to precede identity [68]. However, as Hay [69] points out, place attachment is not stable and is regulated by factors such as the length of residence.

The new generation of migrant workers and peasant groups appear to have a certain degree of differentiation, showing the common attributes of both local people and foreigners. Due to being born and then living in the countryside before adulthood, there is place attachment to rural areas. Periods of long-term residence mean that new generations of farmers still maintain a close emotional and functional connection with their villages.

Based on this, the following research hypotheses are proposed:

**Hypothesis 4 (H4).** *SoP has a mediation effect on perceived impacts of rural tourism and willingness to participate in rural tourism activities.*

**Hypothesis 5 (H5).** *There is a direct positive relationship between place identity and place attachment.*

*2.3. Structural Relation Model*

The main framework of the theoretical model of the new generation of farmers participating in rural tourism development was created using the HOE model: perception leads to sense of place, which leads to willingness. According to the theory of social representation, group perception is included in the analysis of perception. Furthermore, according to the theoretical analysis of sense of place, the sense of place is separated into place identity and place attachment, and the correlation between local identity and local attachment is constructed. Based on the theoretical analysis and research hypotheses proposed above, a structural equation model describing the influence of perceptions of the impact of rural tourism on the willingness to participate in rural tourism activities of the new generation of farmers is constructed. The model is a structural equation model with causality. The model consists of 4 basic dimensions (latent variables) and 12 observation variables. Three types of perception variables are incorporated: economic perception, socio-cultural perception, and environmental perception. The three levels of influence are introduced as independent variables of emotional aspects, with place identity and place attachment as intermediaries. The willingness to participate in rural tourism activities by the new generation of farmers (willingness stage variable) is the dependent variable. The conceptual diagram of the theory and hypotheses of this analysis are shown in Figure 2.

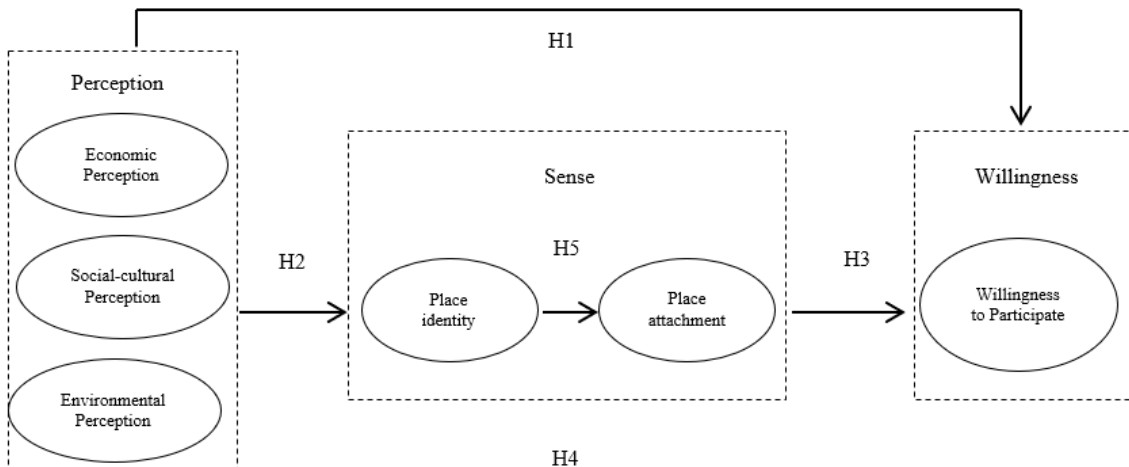

**Figure 2.** Theoretical model and research hypotheses.

## 3. Materials and Methods

### 3.1. Research Area Overview

Yanling County in China was chosen as the ideal area for this study. Yanling County is affiliated with Xuchang City, Henan Province, in the central part of China. As one of the best areas for rural tourism in China, it is the "national leisure, agriculture, and rural tourism demonstration county", "national all-for-one tourism destinations". Yanling County has a long history of flower planting, with the county's flowers and plants covering an area measuring 650,000 mu (Mu is a Chinese unit, which is about 667 square meters), containing more than 2400 varieties. It has been named the "hometown of Chinese flowers and trees" and "China's flower and wood capital". Based on the flower industry, rural tourism in Yanling County has developed rapidly. In 2018, Yanling County built more than 120 rural tourism scenic spots and more than 190,000 people engaged in rural tourism.

Although rural tourism in Yanling County has developed well, after more than 20 years of development, it faces the problem of aging business owners and the intergenerational replacement of rural tourism. A large number of second-generation farmers choose to stay in large cities such as Zhengzhou, which further exacerbate the aging of the countryside. Xuchang City, as a major city with a population outflow in the central region, provides a large number of migrant workers for large cities. By taking Yanling County as the focus of this research, it can provide a fresh sample with which to explore the willingness of the new generation of farmers to participate in rural tourism development.

### 3.2. Questionnaire Design and Variable Measurement

The questionnaire comprised three main items (Appendix A). All of the survey items were developed through a review of the available literature, as well as through the actual situation of rural tourism in China. All items were measured using a 5-point Likert scale (1 = strongly disagree, 5 = strongly agree), which was used to examine the respondents' level of agreement with various statements appearing in the questionnaire. Although the survey statements were primarily derived from the English language literature, we adjusted for this according to the situation of rural tourism in China. The survey statements are listed in Table 1.

The first items aimed to measure perceived rural tourism impacts (P). The survey statements were derived from previous studies on the residents' perceived rural tourism impacts, as well as the broader literature on perceived impacts on the community or of sightseeing tourism [70,71]. Perceived economic impacts (PE) were measured by four items derived from previous research [9,11,43]. Perceived socio-cultural impacts (PS) were measured by three measurement items [6,50,72,73]. Four items were used to evaluate perceived environmental impacts (PEI) (based on those used by Šegota et al. [44] and Hanafiah et al. [74]). In addition, as the new generation of farmers obtain social information more easily,

individual perceptions can be aggregated through the internet to form group perception, which can affect the perception of the new generation of farmers. According to the theory of social representation, group perception is increased. The use of measurement items from different sources helps mitigate the problem of common methodological differences between perceived economic, socio-cultural, and environmental impacts [75].

The second section of the questionnaire captured the new generation of farmers' SoP. Physical place and functional qualities influence the degree of identity towards, and attachment to, a place as a platform for activities and social interaction [27]. Place identity can be extended to objects and places. This refers to the process of individuals or groups interacting with local areas to realize socialization. This special socialization involves many complicated processes, involving emotions, perception, and cognition. Through this process, individuals and groups define themselves as members of a particular place. Place attachment focuses on functional attachment to the place, while place identity emphasizes an individual's emotional identity [28]. Place attachment refers to the emotional connection established between a person and a specific place, expressing the psychological state whereby the individual tends to stay in that place and feel comfortable and safe. The literature indicates that place attachment incorporates the emotional comfort that a place brings to the individual [76], a sense of preference compared to other communities [59], and respect towards what the place stands for [77]. Previous studies have shown that place identity is a multifaceted psychological process that reflects the affective, cognitive, and conative (i.e., behavioral) domains of one's attitude [78–80]. Place identity is considered to conform to all the processes outlined in Breakwell's identity process theory (IPT) [81]. In IPT, the identity structure is artificially constructed in two dimensions: the content dimension, which consists of our social (groups to which we belong to) and individual (our values, motives, attitudes, and emotions) identities; and value dimensions, referring to our assessment of each factor in the content dimension, which determines its importance in the identity hierarchy [82]. Based on this, the place identity can be divided into two parts: sense of belonging [79] and particularity of the place [78,80].

In the third section of the questionnaire, the willingness of the new generation of farmers to participate in rural tourism development was measured by three statements using a Likert-type scale (1 = strongly disagree, 5 = strongly agree). As a direct expression of willingness to participate, direct participation in the development of rural tourism involves employment and investment in the development of rural tourism. The first part captures the willingness of the new generation of farmers to participate directly in the development of rural tourism [83], since Yanling County is at the consolidation stages of the destination life cycle [84]. The second part of the questionnaire focuses on the support for the promotion of local rural tourism, which was expressed by recommending others to participate in the development of local rural tourism [85]. In addition, the benefits of promotion of rural tourism to the outside world were conducive to broader social investment and rural tourism participation. Relaying the positive message of rural tourism to others formed the third part and was related to the willingness to participate [10].

**Table 1.** Specific measurement and references in each dimension of the questionnaire.

| Construction | Code | Items | Reference |
|---|---|---|---|
| Perceived economic impacts (PE) | PE1 | I think rural tourism improves the revenue generated in the local economy. | Rivera et al. [9]; Stylidis et al. [11]; Sinclair-Maragh [43] |
| | PE2 | Rural tourism develops employment opportunities in the rural community. | |
| | PE3 | Rural tourism holds great promise for a rural community's economic future. | |
| | PE4 | It is widely believed that the development of rural tourism in rural areas has a positive effect on rural development. | |
| Perceived social-cultural impacts (PS) | PS1 | Rural tourism improves the quality of life in my hometown. | Almeida-García et al. [6]; Abdollahzadeh and Sharifzadeh [50]; Alipour and Kaboudi [72]; Li et al. [73] |
| | PS2 | Rural tourism development provides more recreational opportunities for locals. | |
| | PS3 | Rural tourism improves the image of rural areas. | |
| Perceived environmental impacts (PEI) | PEI1 | Rural tourism provides more parks and other recreational areas for local residents. | Šegota et al. [44]; Hanafiah et al. [74] |
| | PEI2 | Rural tourism improves the appearance (and images) of a rural community's landscape. | |
| | PEI3 | I think rural tourism improves the living environment. | |
| Place identity (PI) | PI1 | I feel my hometown is a part of me. | Kneafsey [78]; Williams et al. [79]; Kyle et al. [80] |
| | PI2 | My hometown is very special to me. | |
| | PI3 | I have a special connection to the people who live in my hometown. | |
| Place attachment (PA) | PA1 | Staying here makes me forget my problems. | Scannell and Gifford [48]; Lee [60]; Ujang [76] |
| | PA2 | Many of my friends/family prefer their hometown over other communities. | |
| | PA3 | I respect what my hometown stands for. | |
| Residents' willingness to participate (WTP) | W1 | I plan to participate or invest in the development of rural tourism. | Zhang and Zhao [10]; Ajzen [83]; Han and Kim [85] |
| | W2 | I would like to recommend others to participate or invest in rural tourism operation. | |
| | W3 | I will relay the positive message of rural tourism to others. | |

### 3.3. Data Collection and Sample Overview

This data was obtained from Yanling County during research on rural tourism management and sustainable development. According to previous research about the new generation of farmers and the new generation of migrant workers, in conjunction with the definition of the new generation of migrant workers by the Household Survey Office of the National Bureau of Statistics, the population group born after January 1, 1980, was selected as the research group [86–88]. The questionnaire was conducted using a one-on-one, face-to-face questionnaire survey. The content of the questionnaire included the status of the second-generation farmer and their family, and the factors affecting perception, SoP, and willingness to participate in rural tourism.

Before the formal investigation, a pilot questionnaire was designed based on the content of items informed by the literature. Afterwards, three doctoral students studying rural tourism referred to the rural tourism situation in Yanling County to revise the expressions of the items to determine the measurement indicators of each variable. Then, two professors specializing in rural tourism and two professors specializing in population mobility revised the questionnaire again, leading to a more complete questionnaire. Before the final questionnaire survey was implemented, 50 pre-investigations of questionnaires were conducted, and based on the pre-survey results, the items that were found to be unclear or ambiguous were revised to form the final questionnaire, which is shown in Appendix A.

The implementation of the formal survey was conducted over two phases. The first questionnaires were implemented from February 8 to February 13, 2018. The members of the research team entered a village in Yanling County to conduct a randomly implemented questionnaire survey with eligible second-generation farmers. In a face-to-face manner, 150 questionnaires were distributed and 143 were collected. The second survey was conducted from February 18 to February 22, 2019, at Yanling railway station and Yanling bus station. In order to avoid overlap with the first implementation of the survey, the first questionnaire was collected. Before commencing the second phase of the questionnaire implementation, respondents were first asked whether they had completed a similar questionnaire before, and if they had, then they were excluded from this second phase. A total of 250 questionnaires were distributed and 194 were collected. A total of 400 questionnaires were distributed across the two phases, and 337 questionnaires were collected in total. After questionnaire screening, 77 invalid samples were excluded, and 263 valid questionnaires were obtained. The effective completion rate of the questionnaire was 78.04%.

A discrepancies test revealed there was no significant differences between the two subsamples. Table 2 presents the descriptive statistics of the sample. Among the respondents, men accounted for 44.49% and women accounted for 55.51%, and in terms of ethnic groups, ethnic minority groups accounted for 16.73% of the sample and the Han ethnicity accounted for 83.27%. In terms of age, respondents aged 26 to 30 years old accounted for 52.47% of the sample. This age group has a certain level of knowledge and work experience and can better provide power for rural tourism. The sample group had a high level of knowledge, among which university educated individuals comprised the largest sub-group by education level (66.92%), followed by those educated to high school level, who accounted for 14.45% of the sample. Although the success of rural tourism has attracted more attention, there are still 40.68% of the respondents' relatives and friends who are not involved in the industry. Families with an annual income between 50,000 and 200,000 Yuan accounted for 57.42% of the sample. The population living in rural areas for 13–18 years accounted for 34.60% of the sample, while the population who had been living in the countryside accounted for 36.50%. The length of time spent in the countryside may have influenced their sense of place and willingness to participate in the development of their hometown. Additionally, 69.20% of respondents had a relatively stable job.

**Table 2.** Characteristics of respondents.

| Variables | N | (%) |
|---|---|---|
| Gender | | |
| Male | 117 | 44.49 |
| Female | 146 | 55.51 |
| Nationality | | |
| Han | 219 | 83.27 |
| Minority | 44 | 16.73 |
| Age | | |
| Less than 20 | 11 | 4.18 |
| 20–25 | 67 | 25.48 |
| 26–30 | 138 | 52.47 |
| 31–35 | 20 | 7.60 |
| 36–39 | 27 | 10.26 |
| Maximum education level reached | | |
| Less than junior high school | 6 | 2.28 |
| Junior high school | 17 | 6.46 |
| High school graduate | 38 | 14.45 |
| Some college | 176 | 66.92 |
| Graduate student | 25 | 9.51 |
| Family and friends involvement in rural tourism * | | |
| I engaged or invested in | 15 | 5.70 |
| My parents or relatives engaged or invested in | 23 | 8.75 |
| My friends engaged or invested in | 66 | 25.10 |
| My relatives and friends are employed by rural tourism enterprises | 103 | 39.16 |
| None of them are involved | 107 | 40.68 |
| Annual household income | | |
| Less than 50,000 Yuan | 47 | 17.87 |
| 50,000–100,000 Yuan | 70 | 26.62 |
| 100,001–200,000 Yuan | 81 | 30.80 |
| 200,001–300,000 Yuan | 33 | 12.55 |
| 300,000 Yuan or more | 32 | 12.17 |
| Length of residence in rural area | | |
| 1–6 years | 21 | 7.98 |
| 7–12 years | 19 | 7.22 |
| 13–18 years | 91 | 34.60 |
| 19–25 years | 36 | 13.69 |
| always | 96 | 36.50 |
| Employment | | |
| Part time | 26 | 9.89 |
| Full time | 182 | 69.20 |
| Self-employed | 45 | 17.11 |
| Not currently employment | 10 | 3.8 |

* Note: This option allowed for multiple selections.

## 3.4. Methodology

The structural equation model is an ideal model to evaluate the relationship between perceived impacts of rural tourism and willingness to participate in rural tourism. Structural equation modeling is a linear model that evaluates a set of observed variables with fewer unobserved variables [11]. The structural equation model consists of measurement model and structure model. The latent variable factor measurement model describes the latent variables $\xi$ and $\eta$, and observes the relationship between the variables $X$ and $Y$.

$$Y = \Lambda_Y \eta + \varepsilon \tag{1}$$

$$X = \Lambda_X \xi + \delta \tag{2}$$

where $X$ is a vector composed of exogenous observation variables, $\eta$ represents the endogenous latent variable, and $\xi$ represents the exogenous latent variable; $\Lambda_Y$ represents endogenous observed variables, born from the latent variable factor loading matrix, which represents the endogenous latent variable $\eta$ and the observation of the relationship between the variable $Y$. Here, $\Lambda_X$ represents observation of exogenous variables in the factor loading matrix of the exogenous latent variables; this means an exogenous latent variable $\xi$ is deduced, and the observation of the relationship between variable $X$, $\varepsilon$, and $\delta$ is the residual matrix of the measurement equation.

The structural model illustrates the relationship between exogenous latent variables and endogenous latent variables. This relationship is expressed graphically to form a path diagram.

$$\eta \;=\; B\eta + \Gamma\xi + \gamma \tag{3}$$

where $B$ is the structure coefficient matrix, which represents the structure model of endogenous latent variable $\eta$ and the composition of the factors that influence each other; $\Gamma$ is the structure coefficient matrix, which represents exogenous latent variable $\xi$ in the model of the factors' influence on endogenous latent variable $\eta$; $\gamma$ is the residual matrix of the structural model.

The analysis of the sample data and the study hypotheses were based on a two-stage analysis method [10]. Descriptive statistical analysis of the samples was first performed using Stata 15.0 and the reliability and validity of the scales were examined to ensure data quality [89,90]. AMOS24.0 software (AMOS software is analytical software from IBM, Corporate headquarters: 1 New Orchard Road Armonk, New York 10504-1722 United States US: 914-499-1900) was used to analyze the structural equations and study the covariation between variables.

### 3.5. Verification of Common Method Bias

This study collected the subjective views of respondents using self-reported scales, and thus it was possible that common method bias may have been present due to single sample sources, resulting in the expansion or reduction of correlation between constructs. This study avoided the bias caused by common method variation by utilizing a pre-existing questionnaire design and post-testing. During the pilot testing questionnaire section, the error due to common method variation from the design of the measurement item and the arrangement of the questionnaire content was avoided. At the same time, in order to resolve any concerns amongst the respondents, the respondents were first reassured before completing the questionnaire that the information obtained would be kept confidential. In the post-test section, this study examined common method variation between study variables using Harman's one-factor test based on the recommendations of Podsakoff et al. [75].

### 3.6. Bias Reliability and Validity Analysis

The analysis of reliability and validity of the data source was examined. The reliability analysis of the variables was estimated using Cronbach's alpha coefficient [91], using SPSS 24.0 software (SPSS 24.0 software is analytical software from IBM, Corporate headquarters:1 New Orchard Road Armonk, New York 10504-1722 United States US: 914-499-1900). The Cronbach's alpha coefficient for each latent variable was found to be between 0.823 and 0.889, which is greater than 0.7 [92,93], indicating that the data used in this study had good reliability and internal consistency.

The data validity was analyzed based on three aspects: content validity, convergence validity, and discriminant validity [10,90]. Firstly, the data collected in this study was based on a previous research scale, which had been adjusted by experts and included pre-investigation, meaning that it had certain content validity. Secondly, the convergence validity was measured by testing the combination reliability (CR) and the average variation extraction (AVE). As shown in Table 3, the average variation extract (AVE) was between 0.620 and 0.746, the reliability (CR) of each construct was between 0.830 and 0.898, and since both were greater than 0.7, this indicates that the four latent variables had good convergence efficiency. Finally, based on the method of Hair et al. [92], the square roots of the AVE

values of each construct in this study were larger than the correlation coefficient between them and other constructs, indicating that the six latent variables had higher discriminant validity.

**Table 3.** The convergent validity and the difference validity of the variables.

| Variables | AVE | CR | P | PI | PA | WTP |
|---|---|---|---|---|---|---|
| P | 0.746 | 0.898 | 0.864 | - | - | - |
| PI | 0.729 | 0.890 | 0.439 | 0.854 | - | - |
| PA | 0.620 | 0.830 | 0.365 | 0.590 | 0.788 | - |
| WTP | 0.634 | 0.837 | 0.461 | 0.355 | 0.469 | 0.797 |

Notes: The diagonal number indicates the square root of the average variation extract (AVE), and the remaining values are the coefficient of variation between constructs.

In summary, the research model had good reliability and validity, and the inherent quality of the model was ideal. The measurement index can effectively reflect the potential characteristics of the common factor construct. Therefore, further model measurements can be made.

## 4. Results

### 4.1. The Measurement Model

According to Kline [93], the maximum likelihood estimation method can be used to test the hypotheses. Six classic model-fit statistics were used in this study: $\chi^2$/degrees-of-freedom ($\chi^2$/df), goodness of fit index (GFI), standardized root mean square error of approximation (RMSEA), comparative fit index (CFI), incremental fit index (IFI), and normed fit index (NFI). The analysis results found that $\chi^2$/df = 2.732, GFI = 0.925, RMSEA = 0.08, NFI = 0.931, CFI = 0.955, and IFI = 0.955, all of which reached the recommended standard value range [10,93], indicating that the model and the data are adequate. The suggested minimum cut-off values and the modified observed values are presented in Table 4.

**Table 4.** Goodness-of-fit test statistics.

| Fit Index | Absolute Fitting Index | | | Value-Added Fitting Index | | |
|---|---|---|---|---|---|---|
| | $\chi^2$/df | GFI | RMSEA | NFI | CFI | IFI |
| Suggested value | <3 | ≥0.9 | ≤0.08 | ≥0.9 | ≥0.9 | ≥0.9 |
| Observed value | 2.732 | 0.925 | 0.08 | 0.931 | 0.955 | 0.955 |
| Conclusion | Accepted | Good fit | Good fit | Good fit | Good fit | Good fit |

Notes: Abbreviations: $\chi^2$/degrees-of-freedom = $\chi^2$/df; goodness of fit index = GFI; standardized root mean square error of approximation = RMSEA; comparative fit index = CFI; incremental fit index = IFI; normed fit index = NFI.

### 4.2. Subsection Tests of Hypotheses

The structural equation model (SEM) was used to test the hypothetical relationship between the latent variables. AMOS24.0 was used to obtain the path analysis diagram shown in Figure 3, which shows the standardized path coefficient and the *p*-value describing the assumed relationship between place influence perception (P), place identity (PI), place attachment (PA), and willingness to participate (WTP) in rural tourism. Table 5 shows the corresponding path coefficient analysis results. It can be seen from the path coefficient that perceived impacts of rural tourism (P) had a positive impact on willingness to participate in rural tourism (WTP) ($\beta_1 = 0.350$, $P < 0.01$), thus hypothesis 1 was verified; perceived impacts of rural tourism (P) had a positive impact on place identity (PI) ($\beta_2 = 0.457$, $P < 0.01$), thus hypothesis 2 was validated; place attachment (PA) had a positive impact on willingness to participate in rural tourism (WTP) ($\beta_3 = 0.392$, $P < 0.01$), thus hypothesis 3 was verified; place identity (PI) had a positive impact on place attachment (PI) ($\beta_4 = 0.714$, $P < 0.01$). Therefore, all the hypotheses were verified, and the actual analysis results of the model were consistent with the model assumptions.

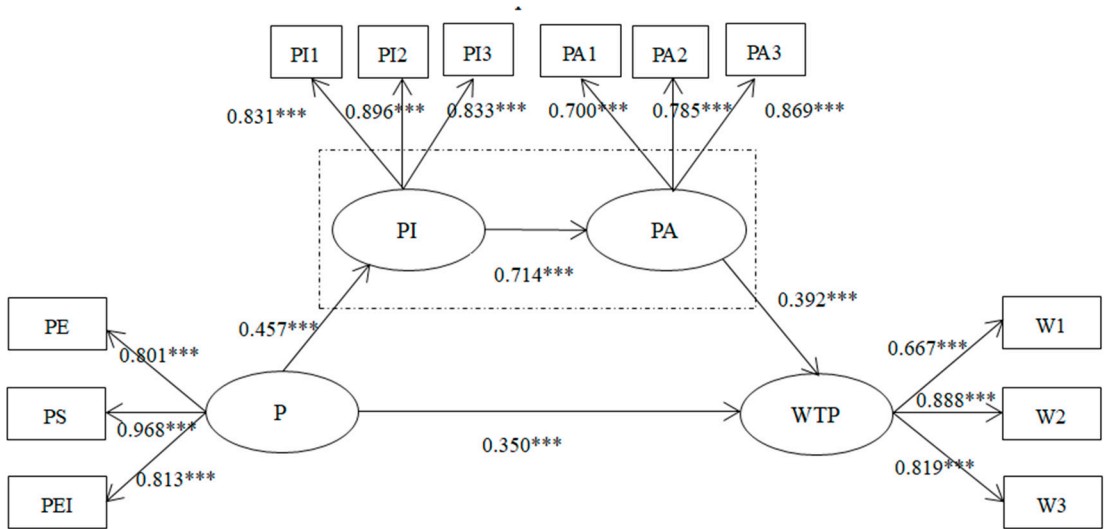

**Figure 3.** The results of the structural equation model (SEM) analysis.

**Table 5.** Path analysis results.

| Paths | Standaredised-Estimated | S.E. [a] | C.R. [b] | Results |
|---|---|---|---|---|
| $P \rightarrow WTP\,(\beta_1)$ | 0.350 *** | 0.096 | 4.909 | Support |
| $P \rightarrow PI\,(\beta_2)$ | 0.457 *** | 0.074 | 7.257 | Support |
| $PA \rightarrow WTP\,(\beta_3)$ | 0.392 *** | 0.068 | 5.256 | Support |
| $PI \rightarrow PA\,(\beta_4)$ | 0.714 *** | 0.088 | 10.232 | Support |

Notes: *** $p < 0.001$; S.E. [a] represents the estimate of the standard error of the covariance; C.R. [b] is obtained by dividing the covariance estimate by its standard error.

### 4.3. Test of Mediation Effect

The method provided by Baron and Kenny [94] was used to examine the mediation effects. The unmediated model can be seen in Figure 4a, which shows that variable X affects variable Y directly. Path C indicates the total effect. After adding the mediation variable Z, the mediation model is shown in Figure 4b. Path c′ represents the directed effect. Path c′ equals zero if variable X no longer affects Y. The process was called complete mediation. When c′ was not equal to zero, it was called the partial mediation. After controlling Z (c–c′), the effect of X on Y decreases, which was called the indirect effect or intermediate quantity. The test for the mediating effect significance is equivalent to the test of the null hypothesis [95].

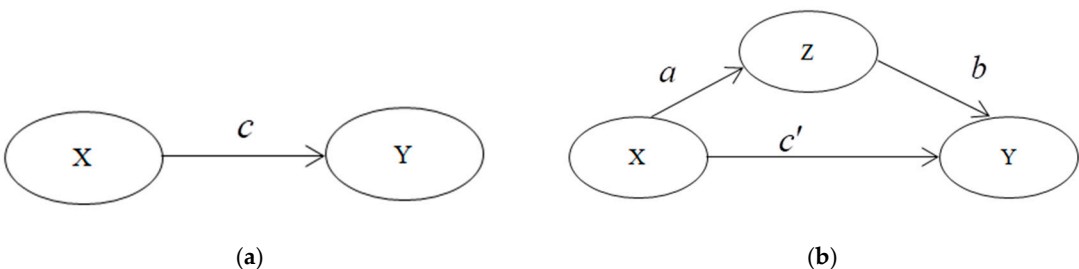

**Figure 4.** Baron and Kenny meditation model: (**a**) unmediated model; (**b**) meditated model.

Using the method above, the full regression models defined by Models (1)–(7) were analyzed with the results shown in Table 6. Stata 15.0 was used to estimate the full regression parameter. The first regression model shows that the coefficient between perceived impacts of rural tourism (P) and willingness to participate (WTP) is 0.619 and $P < 0.01$, indicating that the influence coefficient of P on WTP was significant without a mediating variable. Model (2) shows that perceived impacts of rural

tourism (P) had a direct impact on place identity (PI), with an impact coefficient of 0.521, which is significant at the 1% significance level. Model (3) shows that place identity (PI) has a direct impact on place attachment (PA), with an impact coefficient of 0.622, which was significant at the 1% significance level. Model (4) shows that place attachment (PA) had a direct impact on the willingness to participate, with an impact coefficient of 0.473, which was significant at the 1% significance level. Model (5) shows the combined influence of perceived impacts of rural tourism (P) and place identity (PI) on place attachment (PA), where place identity (PI) mediates rural tourism impact perception (P) and place attachment (PA). The effect, and thus hypothesis 5, was verified. Model (6) shows the combined effects of place identity (PI) and place attachment (PA) on willingness to participate (WTP). Place attachment (PA) has a mediating effect on place attachment (PA) and willingness to participate (WTP), which is shown in the results from Model (7). This shows the combined influence of perceived impacts of rural tourism (P) and sense of place (SoP) on willingness to participate (WTP). The direct impact of rural tourism impact perception (P) on participation willingness (WTP) is greater than the indirect impact (0.619, 0.173 = 0.619 − 0.446), and the model results show that the influence of sense of place (SoP) on rural tourism impact perception (P) and willingness to participate (WTP) have a mediating effect; thus, hypothesis 4 was verified.

**Table 6.** Tests for mediation effects by regression analysis.

|  | Regression Model | $b$ | $c$ | $d$ | $c'$ | $R^2$ |
|---|---|---|---|---|---|---|
| (1) | WTP $= \alpha + c\text{P} + \varepsilon$ | - | 0.619 *** | - | - | 0.212 |
| (2) | PI $= \alpha + c\text{P} + \varepsilon$ | - | 0.521 *** | - | - | 0.193 |
| (3) | PA $= \alpha + d\text{PI} + \varepsilon$ | - | - | 0.662 *** | - | 0.348 |
| (4) | WTP $= \alpha + b\text{PA} + \varepsilon$ | 0.473 *** | - | - | - | 0.220 |
| (5) | PA $= \alpha + b\text{P} + d\text{PI} + \varepsilon$ | 0.175 ** | - | 0.597 *** | - | 0.362 |
| (6) | WTP $= \alpha + d\text{PI} + b\text{PA} + \varepsilon$ | 0.402 *** | - | 0.137 * | - | 0.229 |
| (7) | WTP $= \alpha + c'\text{P} + d\text{PI} + b\text{PA} + \varepsilon$ | 0.347 *** | - | 0.00743 | 0.446 *** | 0.316 |

Notes: *** $p < 0.001$, ** $p < 0.005$, * $p < 0.01$.

Further, the mediation effect was examined using the method proposed by Sobel [95]. Table 7 shows the test results of the mediation effect obtained by AMOS 24.0. The test results show that the overall impact of perceived impacts of rural tourism (P) on willingness to participate (WTP) is 0.479, for which the direct influence coefficient is 0.350 and indirect influence coefficient is 0.129. The influence coefficient is 0.128, and as the influence of perceived impacts of rural tourism (P) on willingness to participate (WTP) is adjusted to some extent by sense of place (SoP), there is a partial mediation effect, which further verifies hypothesis 4. The effect of perceived impacts of rural tourism (P) on place attachment (PA) is 0.784, for which the direct impact coefficient is 0.457 and the indirect impact coefficient is 0.327; thus, perceived impacts of rural tourism (P) impacts on place attachment (PA) to some extent, which is regulated by place identity (PI), and there is a partial mediation effect. In addition, the overall impact of place identity (PI) on willingness to participate (WTP) is 0.280, for which the indirect impact coefficient is 0.280, and there is no direct impact on local influence (PI) affecting willingness to participate (WTP) through place attachment (PA). There is a complete mediating effect, which further validates hypothesis 5. In general, the results of the mediation effect show that there is one complete mediation effect and two partial mediation effects in the influence path constructed in this paper. Even though the indirect influence coefficient of perceived impacts of rural tourism on willingness to participate is small at only 0.129, the coefficients from P to PI (0.457) and from PI to PA (0.714) are large, which help to construct the indirect impact path from P to WTP.

**Table 7.** Tests for mediation effects by path analysis.

|  | Total Effects | Direct Effects | Indirect Effects | Results |
| --- | --- | --- | --- | --- |
| P → WTP | 0.479 *** | 0.350 *** | 0.129 *** | Partial mediation |
| P → PI | 0.457 *** | 0.457 *** | - | No mediation |
| PI → PA | 0.714 *** | 0.714 *** | - | No mediation |
| P → PA | 0.784 *** | 0.457 *** | 0.327 *** | Partial mediation |
| PA → WTP | 0.392 *** | 0.392 *** | - | No mediation |
| PI → WTP | 0.280 *** | - | 0.280 *** | Full mediation |

Notes: *** $p < 0.001$.

## 5. Discussion

This study has examined perceived impacts of rural tourism, sense of place, and willingness to participate in rural tourism in China. Our results reveal that when perceived impacts of rural tourism are the only independent variable, this variable has a significant direct positive impact on participation willingness. When sense of place is introduced into the model, the direct impact of the variable perceived impacts of rural tourism on participation will decrease. Our results show that the influence of perceived impacts of rural tourism on participation are not only direct but also indirect, with SoP playing a mediating role. The coefficient of direct influence (0.350) is greater than that of indirect influence (0.129), indicating that although there is an indirect influence path, the influence is still dominated by direct influence. In addition, considering local attachment and local identity as two separate dimensions is necessary. Place identity plays an intermediary role in the influence of perceived impacts of rural tourism and place attachment, and place attachment plays a mediating role between place identity and willingness to participate.

Despite the plethora of research that has already been conducted in this field, research on the participation of the new generation of farmers in rural tourism development continues. This kind of exploration is especially important for developing countries such as China, where rural tourism is not developed. From the perspective of sustainable development, by improving the media publicity of successful rural tourism cases and improving rural infrastructure, this is conducive to improving the positive cognition and confidence of the new generation of farmers, leading them to return home and start their own businesses. This will stimulate their local attachment to and identity with the countryside, as well as improve their willingness to return home to engage or invest in rural tourism. The findings of the present study have implications for the development of related theories and management of rural tourism in the future.

### 5.1. Implications for Future Research

The current research provides a feasible path for further study of the new generation of farmers participating in the development of rural tourism in China. First of all, although some studies have already conducted research on intergenerational participation in rural tourism, this study has utilized the hierarchy of effects (HOE) model which has enriched the research framework for the examination of the sustainable development of rural tourism in China. Secondly, sense of place (SOP) is separated into place identity (PI) and place attachment (PA), and in the model analysis in this study, the mediating effects of sense of place (SOP), place identity (PI), and place attachment (PA) have each been discussed. Consequently, this study has verified the role of SoP in intergenerational participation in rural tourism. Future research should pay more attention to the impact of participation and resident satisfaction on the sustainable development of rural tourism in China.

This study has some limitations that must be acknowledged, although these do provide opportunities for future research. First, the data for this study was obtained from China, which may reduce the applicability of these study findings to other developing countries due to the different characteristics of the inter-country population and rural tourism development in China compared to these other studies. Secondly, this study used cross-sectional data, which meant that any dynamic

changes in the influence perception and participation willingness of rural tourism over the lifecycle of tourism could not be captured. The choice of Yanling County in this study means the results only represent the willingness of respondents to participate in rural tourism destinations over the development period in this location. Thirdly, due to different resource endowments, the development modes of rural tourism are quite different. Farmers' access and investment levels also differ greatly due to different development modes. This study only focused on types of rural tourism based on farmhouse and leisure farm management, and so other types of rural tourism should be considered in future works.

*5.2. Implications for Pratice*

This study may have two important practical implications. First, the willingness of the new generation of farmers to participate in rural tourism is not only directly affected by their perception of rural tourism, but also indirectly influenced by their local feelings. The weakening SoP may be the root cause of rural tourism and even the weakening of rural industries, which could also be an explanation for the intergenerational replacement problem faced by rural tourism. If the local government promotes functional and emotional propaganda related to the SoP in the process of guiding the second generation of farmers to return home, this will help to promote the new generation of farmers to return to the countryside to engage in the development of rural tourism.

Secondly, the introduction of group perception into the model analysis in this study supplements the overlooked group influence in the social exchange theory and helps to provide an understanding of the characteristics of the second-generation group as a social group susceptible to group perception. The common psychological characteristics of the new generation of farmers are important factors affecting their return to home business, and become an important path to business policies that will encourage these individuals to return home.

The conclusion of this paper provides a good reference for the Chinese government (or other developing countries) to implement a policy of sustainable development and rejuvenation of rural tourism. In rural China, under the context of aging of rural tourism practitioners, in order to implement sustainable rural tourism and rural tourism service promotion, the following steps should be taken: firstly, local government should strengthen the positive impact of rural tourism, which will catch second-generation farmers' attention; secondly, the government should consummate rural infrastructure, excavate the rural culture connotation, and awaken and strengthen second-generation farmers' sense of place. In the end, preferential policies for investment in business are necessary to translate willingness to participate into actual investment or business.

## 6. Conclusions

This study uses survey data collected at the rural level to examine the intergenerational replacement of rural tourism management, and used the hierarchy of effects (HOE) model to design a three-stage model based on "cognition-emotion-willingness", which takes the positive impact of rural tourism development as an external stimulus. These variables mainly examined the influence of the rural tourism perception of the new generation farmers on their willingness to participate in rural tourism. Social representation theory (SRT) was adopted to incorporate group cognition into the residents' perceptions of the impact of rural tourism, and the research on the influence of residents' perceptions on willingness was improved. This study takes research on the willingness of the new generation of farmers to return home and argues that SoP can regulate the perception and willingness to participate in rural tourism. In contrast to the results of previous studies focused on SoP, this study further validates the mediating role of place attachment (PA) in place identity (PI) and willingness to participate. The research results are helpful in providing an understanding of returning entrepreneurship or investment phenomenon in the process of rural tourism development, and are conducive to the theoretical development of intergenerational replacement in the sustainable development of rural tourism.

Research on the willingness to participate in rural tourism by the new generation of farmers has been influenced by many aspects. The results show that the influence of SoP on the rural tourism industry's willingness to participate in rural tourism activities is significant, and an improvement in SoP will lead to more rural second-generation groups returning to their hometowns to participate in rural tourism development. It can be seen from the decomposition of sense of place that local identity and local attachment together constitute the mediating variable for the second generation of farmers to participate in rural tourism development. Local identity will affect local attachment, and thus affect participation intention. Local identity and local attachment are equally important to the willingness to participate.

This current research has made a theoretical contribution to China's ongoing efforts to encourage talented local residents to return to their hometowns and promote rural revitalization. In the context of the impact of perceptions of the impact of rural tourism on willingness to participate, future research in this area should incorporate SoP into any model design considerations. Future research may also consider other intermediaries, such as the degree of participation and resident satisfaction.

**Author Contributions:** B.L. and Z.M. conceived and designed the article framework and the questionnaire. B.L. collected and analyzed the data and wrote this paper. B.L. and Z.Z. made modifications. All authors have read and agreed to the published version of the manuscript.

**Funding:** This research was supported by the National Natural Science Foundation of China (grant number 71573259).

**Conflicts of Interest:** The authors declare no conflict of interest.

## Appendix A  Questionnaire

*Section A: Perceived Impacts of Rural Tourism*

**Table A1.** **Do you agree or disagree with the following statements about Yanling County? (1 = strongly disagree, 2 = disagree, 3 = neither agree nor disagree, 4 = agree, 5 = strongly agree.** Please select one of the following options to tick.).

| Perceived Impacts of Rural Tourism | Strongly Disagree | Disagree | Neither Agree nor Disagree | Agree | Strongly Agree |
|---|---|---|---|---|---|
| 1. I think rural tourism improves the revenue generated in the local economy. | 1 | 2 | 3 | 4 | 5 |
| 2. Rural tourism develops employment opportunities in the rural community. | 1 | 2 | 3 | 4 | 5 |
| 3. Rural tourism holds great promise for a rural community's economic future. | 1 | 2 | 3 | 4 | 5 |
| 4. It is widely believed that the development of rural tourism in rural areas has a positive effect on rural development. | 1 | 2 | 3 | 4 | 5 |
| 5. Rural tourism improves the quality of life in my hometown. | 1 | 2 | 3 | 4 | 5 |
| 6. Rural tourism development provides more recreational opportunities for locals. | 1 | 2 | 3 | 4 | 5 |
| 7. Rural tourism improves the image of rural areas. | 1 | 2 | 3 | 4 | 5 |
| 8. Rural tourism provides more parks and other recreational areas for local residents. | 1 | 2 | 3 | 4 | 5 |
| 9. Rural tourism improves the appearance (and images) of a rural community's landscape. | 1 | 2 | 3 | 4 | 5 |
| 10. I think rural tourism improves the living environment. | 1 | 2 | 3 | 4 | 5 |

*Section B: Sense of Place*

**Table A2.** **Do you agree or disagree with the following statements about your hometown? (1 = strongly disagree, 2 = disagree, 3 = neither agree nor disagree, 4 = agree, 5 = strongly agree.** Please select one of the following options to tick.).

| Sense of Place | Strongly Disagree | Disagree | Neither Agree Nor Disagree | Agree | Strongly Agree |
|---|---|---|---|---|---|
| 11. I feel my hometown is a part of me. | 1 | 2 | 3 | 4 | 5 |
| 12. My hometown is very special to me. | 1 | 2 | 3 | 4 | 5 |
| 13. I have a special connection to the people who live in my hometown. | 1 | 2 | 3 | 4 | 5 |
| 14. Staying here makes me forget my problems. | 1 | 2 | 3 | 4 | 5 |
| 15. Many of my friends/family prefer their hometown over other communities. | 1 | 2 | 3 | 4 | 5 |
| 16. I respect what my hometown stands for. | 1 | 2 | 3 | 4 | 5 |

*Section C: Willingness to Participate in Rural Tourism Development*

**Table A3.** **Do you agree or disagree with the following statements? (1 = strongly disagree, 2 = disagree, 3 = neither agree nor disagree, 4 = agree, 5 =strongly agree.** Please select one of the following options to tick.).

| Willingness to Participate in Rural Tourism Development | Strongly Disagree | Disagree | Neither Agree nor Disagree | Agree | Strongly Agree |
|---|---|---|---|---|---|
| 17. I plan to participate or invest in the development of rural tourism. | 1 | 2 | 3 | 4 | 5 |
| 18. I would like to recommend others to participate or invest in rural tourism operation. | 1 | 2 | 3 | 4 | 5 |
| 19. I will relay the positive message of rural tourism to others. | 1 | 2 | 3 | 4 | 5 |

*Section D: Personal and Family Status*

About you (Please tick the appropriate box)

20　What is your gender?

□ Male □ Female

21　What is your nation?

□ Han nationality □ Minority

22　How old are you?

□ Less than 20 □ 20–25 □ 26–30 □ 31–35 □ 36–39

23　What is your maximum education level reached?

□ Less than junior high school

□ Junior high school

□ High school graduate

□ Some college

□ Graduate student

24　Do your relatives or friends participate in rural tourism? (This option allowed for multiple selections.)

□ I engaged or invested in

□ My parents or relatives engaged or invested in

□ My friends engaged or invested in

□ My relatives and friends are employed by rural tourism enterprises

□ None of them are involved

25 What is your family's annual income?

□ Less than 50,000 Yuan

□ 50,000–100,000 Yuan

□ 100,001–200,000 Yuan

□ 200,001–300,000 Yuan

□ 300,000 Yuan or more

26 How long have you lived in rural area?

□ 1–6 years □ 7–12 years □ 13–18 years □ 19–25 years □ always

27 What is your current employment status?

□ Part time □ Full time □ Self-employed □ Not currently employment

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
