# Peer review of "Willingness of the New Generation of Farmers to Participate in Rural Tourism: The Role of Perceived Impacts and Sense of Place"

_sustainability, doi:10.3390/su12030766_

Round 1

Reviewer 1 Report

The topic of this paper is very actual and interesting nowadays. This paper utilizes an analysis model of perceived rural tourism impacts, sense of place and willingness to participate by the new generation of farmers, where the sense of place is decomposed into the two independent dimensions of place identification and place attachment. Author/s examines the willingness to participate in rural tourism development by the new generation of farmers using a structural equation model.

Although my overall assessment of the paper is positive, I would suggest the author(s) to do a minor revision of this paper before publishing. Below are the major issues that I would encourage you to consider in further improving this paper:

The title is not fully clear it is too general An introduction is too wide. The introduction needs more work. There should not be a description of the case study in the introduction. Most of the introduction discusses the China countryside. This paper should not be about the 'China countryside' but rather about a concept. What is the concept of the paper? Is it "residents' willingness to participate.,.." ? I think so. To make a conceptual contribution to the literature, the introduction should be about the topic of the paper, not the location. Otherwise, where is the new knowledge? The information about the China countryside should come later in the description of the case study, usually located at the beginning of the methods section. The introduction needs to 'hook' in the reader with an interest in a concept, not a specific place. The China countryside just happens to be the laboratory where you are studying the concept. That's the way the introduction should appear to readers. So there is more work to be done on the introduction so that it sets the conceptual tone, not a description of the locality. It should be on no more than one page. In introduction, author/s should define the research problem the motive for addressing the problem and structure of the paper without literature background. Many of these things are missing in the present paper and therefore it has to be expanded. 

Author Response

Point 1: The title is not fully clear it is too general

Response 1: Thank you very much for your valuable comments! After rethinking about the content of the article, we found that the original topic was really not clear and was too general. We modified the topic and revised it as: Willingness of the new generation of farmers to participate in rural tourism: the role of perceived impacts and sense of place.

Point 2: The introduction needs more work. There should not be a description of the case study in the introduction. Most of the introduction discusses the China countryside. This paper should not be about the 'China countryside' but rather about a concept. What is the concept of the paper? Is it "residents' willingness to participate.,.." ? I think so. To make a conceptual contribution to the literature, the introduction should be about the topic of the paper, not the location. Otherwise, where is the new knowledge? The information about the China countryside should come later in the description of the case study, usually located at the beginning of the methods section. The introduction needs to 'hook' in the reader with an interest in a concept, not a specific place. The China countryside just happens to be the laboratory where you are studying the concept. That's the way the introduction should appear to readers. So there is more work to be done on the introduction so that it sets the conceptual tone, not a description of the locality. It should be on no more than one page. In introduction, author/s should define the research problem the motive for addressing the problem and structure of the paper without literature background. Many of these things are missing in the present paper and therefore it has to be expanded.  

Response 2: Thank you very much for your valuable comments! The original introduction used many paragraphs to write the background of the development of rural tourism in China, but did not specify the research problem, the motivation to solve the problem and the structure of the paper. After communication with other authors of this paper, we accepted the revision opinions, deleted the background part, and rewrote the introduction. The revised content is shown in the paper.

Reviewer 2 Report

The paper is well designed and developed. prior to be published some minor revisions are necessary:
Lines 135-137: reference needed
line 143: are social exchange theory and Social Representation theory the same?
Discussion section: needs to be enlarged, inserting/moving comments already introduced in the paper (as an example, lines 358-360)

Author Response

Point 1: Lines 135-137: reference needed

Response 1: Thank you very much for your valuable comment! The missing reference of Lines 135-137 in the original text is supplemented.

Point 2: line 143: are social exchange theory and Social Representation theory the same?

Response 2: Thank you very much for your valuable question and comment! Social exchange theory and social Representation theory are two different theories. Social Representation theory holds that in addition to direct experience and individual activities, individuals are also influenced by electronic media and social interaction. In line 143 of the original text, we want to introduce the theory of social representation by discussing the shortcomings of social exchange theory in the research process. Due to the excessive description of existing theories in this paper, the theoretical part has been reduced, and the revised content is shown in the revised version.

Point 3: Discussion section: needs to be enlarged, inserting/moving comments already introduced in the paper (as an example, lines 358-360)

Response 3: Thank you very much for your valuable comment! The comments that have already appeared in this article are expanded and discussed in detail. The revised content is shown in the revised version.

Reviewer 3 Report

Check syntax throughout the text

In the introduction, please present the rationale behind the identified theory and selected methods

Lines 32-37: I suggest synthesising by avoiding any explicit reference to the current political settings.

Line 39: Please explain “lack resources and land”

Lines 43-55: please substantiate the text by referring to the literature

Line 91: the purpose of the study is (not was)

In paragraph 2, there is no need to dedicate much space to known theories/methodologies. This paragraph should provide 1) a literature review of previous research (which is missing) and 2) and analytical framework that brings to the identification of the hypotheses. No research methods should be presented here. Please adjust the paragraph accordingly.

Lines 131-139: citation missing

Paragraph three needs some restructuring, as the paragraph should include the description of research methods and data, in that order. I suggest renaming the paragraph after “materials and methods” or similar. 1) You should describe the theoretical and empirical model, with formulas, otherwise it’s very difficult to understand what you did; 2) research design, including modelling and testing procedure and questionnaire design and 3) data: a) please explain the representativeness of the sample; b) please present summary statistics

Lines 384-387: are you using primary or secondary data? If data are primary, you should just acknowledge project funding at the end of the paper; if data are secondary, you clearly state it and add a relevant citation

The presentation and discussion of results are poor

Please add the policy implications of your study

Author Response

Point 1: Check syntax throughout the text

Response 1: Thank you very much for your valuable comment! There were some non-standard words or grammatical errors in the previous text, so we carefully checked the article and corrected the incorrect grammar.

Point 2: In the introduction, please present the rationale behind the identified theory and selected methods

Response 2: Thank you very much for your valuable comment! After communication with other authors of this paper, we accepted the revision opinions, added theoretical background and description of the principles of the selection method. The revised content is shown in the paper.

Point 3: Lines 32-37: I suggest synthesising by avoiding any explicit reference to the current political settings.

Response 3: Thank you very much for your valuable comment! In the introduction, contents related to the current political settings were deleted. Lines 32-37 in the original text were deleted.

Point 4: Lines 39: Please explain “lack resources and land”。

Response 4: Thank you very much for your valuable comment! "Lack resources and land" in the original text is explained by taking Guangxi and Gansu in China as examples.

Point 5: Lines 43-55: please substantiate the text by referring to the literature

Response 5: Thank you very much for your detail comment! Because the previous introduction devoted too much space to the Chinese countryside, this part was deleted to make the article more focused.

Point 6: Line 91: the purpose of the study is (not was)

Response 6: Thank you very much for your detail comment! Grammatical errors have been corrected and the full text has been checked to avoid similar errors.

Point 7: In paragraph 2, there is no need to dedicate much space to known theories/methodologies. This paragraph should provide 1) a literature review of previous research (which is missing) and 2) and analytical framework that brings to the identification of the hypotheses. No research methods should be presented here. Please adjust the paragraph accordingly.

Response 7: Thank you very much for your valuable comment! We accept the comment and modify from the following aspects:

1) The theory section has been scaled back appropriatelyï¼›

2)A review of relevant literature was added;

3)The analytical framework of identification hypothesis is proposed;

Specific changes are presented in the revised version.

Point 8: Lines 131-139: citation missing

Response 8: Thank you very much for your valuable comment! The missing citations of lines 131-139 in the original paper are supplemented.

Point 9: Paragraph three needs some restructuring, as the paragraph should include the description of research methods and data, in that order. I suggest renaming the paragraph after “materials and methods” or similar. 1) You should describe the theoretical and empirical model, with formulas, otherwise it’s very difficult to understand what you did; 2) research design, including modelling and testing procedure and questionnaire design and 3) data: a) please explain the representativeness of the sample; b) please present summary statistics

Response 9: Thank you very much for your valuable comment! After discussion with other authors, we believe that modification is necessary. We accept the comment and modify from the following aspects:

1) Rename the paragraph “Materials and Methods”ï¼›

2) Model description is added in methodology to explain the method used in the paper with formulas;

3) The questionnaire is in appendix A, and summary statistics and processing processes are presented in the attachment.

  4) Representativeness of the sample:First, Yanling County is a model county for leisure agriculture and rural tourism in China, indicating that this is a representative area. Secondly, a pre-investigation was conducted before the survey, and the selection methods and locations of the survey samples were evaluated. Third, using random sampling to conduct surveys ensures the randomness of the survey samples.

Point 10: Lines 384-387: are you using primary or secondary data? If data are primary, you should just acknowledge project funding at the end of the paper; if data are secondary, you clearly state it and add a relevant citation

Response 10: Thank you very much for your valuable comment! This article use primary data. We removed the introduction to the project from lines 386-389 and only introduced the project funding at the end of the paper.

Point 11: The presentation and discussion of results are poor,

Response 11: Thank you very much for your valuable comment! The discussion and description of the results are added in the fourth part of the article.

Point 12: Please add the policy implications of your study,

Response 12: Thank you very much for your valuable comment! We accept the revision and add policy recommendations in 5.2.

Round 2

Reviewer 3 Report

Dear Authors,

The article has substantially improved and I think is ready for publication; please check for typos thorughout the text

Author Response

Point 1: Please check for typos throughout the text. Response 1: Thank you very much for your comments! The grammar and typos have been carefully checked and revised. The revised paper is shown in the attachment.
